# Automated Quality Assessment of Blind Sweep Obstetric Ultrasound for Improved Diagnosis

**Prasiddha Bhandari**[1]             PRASIDDHA.BHANDARI@NAAMII.ORG.NP
**Kanchan Poudel**[1]               KANCHAN.POUDEL@NAAMII.ORG.NP
**Nishant Luitel**[1]                NISHANT.LUITEL@NAAMII.ORG.NP
**Bishram Acharya**[1]              BISHRAM.ACHARYA@NAAMII.ORG.NP
**Angelina Ghimire**[1]             ANGELINA.GHIMIRE@NAAMII.ORG.NP
**Tyler Wellman**[2]               TYLER.WELLMAN@GEHEALTHCARE.COM
**Kilian Koepsell**[2]              KILIAN.KOEPSELL@GEHEALTHCARE.COM
**Pradeep Raj Regmi**[3]             PRADEEP.REGMI@IOM.EDU.NP
**Bishesh Khanal**[1]               BISHESH.KHANAL@NAAMII.ORG.NP

[1] *Nepal Applied Mathematics and Informatics Institute for research (NAAMII), Nepal*

[2] *GE HealthCare*

[3] *Institute of Medicine (IOM), Tribhuvan University, Nepal*

**Editors:** Accepted for publication at MIDL 2026

## Abstract

Blind Sweep Obstetric Ultrasound (BSOU) enables scalable fetal imaging in low-resource settings by allowing minimally trained operators to acquire standardized sweep videos for automated Artificial Intelligence(AI) interpretation. However, the reliability of such AI systems depends critically on the quality of the acquired sweeps, and little is known about how deviations from the intended protocol affect downstream predictions. In this work, we present a systematic evaluation of BSOU quality and its impact on three key AI tasks: sweep-tag classification, fetal presentation classification, and placenta-location classification. We simulate plausible acquisition deviations, including reversed sweep direction, probe inversion, and incomplete sweeps, to quantify model robustness, and we develop automated quality-assessment models capable of detecting these perturbations. To approximate real-world deployment, we simulate a feedback loop in which flagged sweeps are "re-acquired", showing that such correction improves downstream task performance. Our findings highlight the sensitivity of BSOU-based AI models to acquisition variability and demonstrate that automated quality assessment can play a central role in building reliable, scalable AI-assisted prenatal ultrasound workflows, particularly in low-resource environments.

**Keywords:** Blind sweep ultrasound, quality assessment, obstetric imaging, AI robustness, prenatal care

## 1. Introduction

Obstetrics ultrasound is central to prenatal care, providing critical information on gestational age, fetal development, presentation, and placental position. Reliable clinical decision-making depends on the accurate acquisition and interpretation of these images. Yet in many low-resource settings, shortages of trained sonographers and radiologists, along with limited access to advanced imaging equipment, constrain both the availability and quality of prenatal ultrasound.

The growing availability of low-cost, portable hand-held ultrasound devices presents an opportunity to expand access to obstetrics ultrasound care. To make ultrasound feasible for non-expert operators, simplified acquisition strategies, particularly blind sweeps, have been developed to reduce acquisition complexity while ensuring consistent anatomical coverage. One of the earliest examples is the Imaging the World (ITW) six-sweep protocol (Figure 1), introduced in a teleradiology context where non-experts captured standardized sweep videos for remote expert review (DeStigter et al., 2011). Building on this foundation, more recent initiatives have paired blind-sweep protocols with AI-driven interpretation or prediction, enabling automated extraction of clinically relevant information (van den Heuvel et al., 2019; Pokaprakarn et al., 2022; Self et al., 2022). Several deep-learning-based AI models have been developed that take Blind Sweep Obstetrics Ultrasound (BSOU) videos as input and perform specific tasks such as Gestational Age (GA) estimation (Gomes et al., 2022; Pokaprakarn et al., 2022; Patel et al., 2024), fetal presentation classification (Gomes et al., 2022; Gleed et al., 2023a; Wiśniewski et al., 2025), and placenta location (Gleed et al., 2023b).

Despite the promise of BSOU-based AI models, an open question remains: how sensitive are these models to variations in sweep quality? In real-world deployments, particularly in low-resource settings where operators may have minimal training, it is plausible that sweeps may deviate from the intended acquisition protocol. Potential deviations include reversed or inconsistent sweep directions, incomplete coverage, incorrect probe orientation, or subtle operator-induced motion. Although BSOU protocols are designed to be simple, it is not yet clear how often such issues occur in practice or how strongly they affect downstream AI predictions. Existing AI systems for obstetric BSOU typically assume that input sweeps follow the prescribed protocol, meaning that their robustness to real-world acquisition variability is not well understood.

This motivates the need to systematically study how deviations from protocol affect downstream model performance and to develop automated mechanisms that could detect potential quality issues when they arise. In this work, we focus on quality assessment of BSOU for three tasks: sweep tag prediction, fetal presentation classification, and placenta location classification. We simulate plausible deviations from standard acquisition protocols, such as incomplete sweeps, reversed sequence order, and reduced anatomical coverage, to quantify their impact on task-specific AI models. We further propose automated methods to identify sweeps exhibiting these perturbations.

To approximate a realistic deployment scenario, we designed experiments in which detected low-quality sweeps are assumed to be re-acquired correctly, mimicking real-time operator feedback loops. Our results show that models can be trained to detect several forms of sweep perturbations and that incorporating such detection mechanisms can improve robustness of downstream tasks under simulated acquisition variability.

Our contributions are:

1. a systematic evaluation of how different quality perturbations affect multiple BSOU downstream tasks,

2. the development of automated quality-assessment models for identifying likely sweep deviations, and

3. demonstration of a feedback-and-reacquisition simulation showing improved downstream performance when low-quality sweeps are flagged and re-acquired.

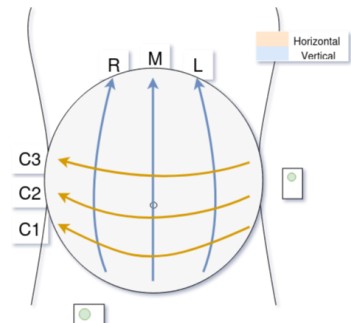

Figure 1: 6-sweep BSOU protocol

## 2. Related works

Quality assessment (QA) plays a crucial role across medical imaging modalities, as both human interpretation and AI models rely heavily on the reliability of the underlying scans. Poor-quality acquisitions, due to motion, incorrect positioning, incomplete anatomical coverage, or operator error, can lead to missed diagnoses, unreliable biomarkers, and degraded model performance. As imaging expands into settings with limited expertise and increased workflow pressures, automated QA has become essential for ensuring that only diagnostically usable data is interpreted or processed downstream.

A large body of work focuses on retrospective detection of non-diagnostic or corrupted images. In MRI, deep learning systems have been developed to detect motion-corrupted slices or non-diagnostic scans, enabling pipelines to exclude problematic data before further analysis (Weaver et al., 2023; Samani et al., 2020). These approaches highlight a broader principle: identifying poor-quality inputs early improves the validity and robustness of any downstream imaging task. Other studies in fetal MRI (Zhang et al., 2020; Xu et al., 2019) show detection of fetal motion or quanitifying fetal movements can allow real-time mitigation of motion artifacts during MRI acquisition.

Retrospective review is also common in ultrasound. For example, (Yaqub et al., 2019) demonstrated improvements in fetal anatomy screening completeness through structured manual review but noted that such processes are slow, resource-intensive, and difficult to scale. This reinforces the need for automated solutions in operator-dependent modalities.

Another class of QA approaches provides feedback during image acquisition. In chest radiography, immediate AI-driven guidance has been shown to improve patient positioning and reduce avoidable exposure, demonstrating how point-of-care feedback can raise imaging standards (Poggenborg et al., 2021). Similar strategies have been applied in mammography, where automated systems emulate expert positioning decisions to reduce repeat exams and support technologists in achieving consistent, high-quality acquisitions (Gupta et al., 2021). These works underscore that QA is not only retrospective but can actively enhance acquisition reliability.

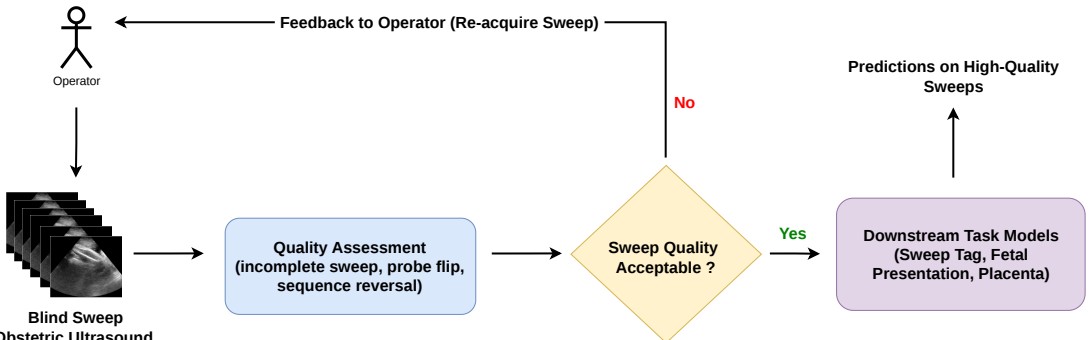

Figure 2: A quality control pipeline in obstetric ultrasound scans, using QA to ensure acceptable sweep quality before processing for downstream tasks.

Beyond explicit QA algorithms, uncertainty estimation has been used as a proxy for quality. In ultrasound segmentation tasks such as echocardiography, (Dahal et al., 2020) showed that uncertainty maps can identify unreliable regions and potentially flag low-quality or out-of-distribution images before segmentation, suggesting a pathway for QA-informed downstream analysis.

Despite the recent progress in AI models for BSOU (Pokaprakarn et al., 2022; Gomes et al., 2022), explicit QA remains largely unexplored. Existing models typically assume that sweeps follow ideal acquisition protocols, leaving open questions about model behavior when sweeps are incomplete, reversed, or otherwise inconsistent with protocol expectations, a scenario that may plausibly arise in low-resource settings or among minimally trained operators. In this work, we address this gap by systematically examining how perturbations in BSOU sweep quality affect downstream AI tasks and by developing automated methods to detect key forms of acquisition deviation. Our goal is to support more reliable, scalable BSOU deployment by ensuring that diagnostic models operate on appropriately acquired input data.

## 3. Methodology

### 3.1. Datasets and Preprocessing

We conducted our study using a large collection of BSOU videos obtained from two clinical sites (Chapel Hill, NC, USA, and Lusaka, Zambia). The dataset contains over 8,000 studies from more than 4,600 pregnant individuals, with each study comprising between 6 and 15 sweeps. Sweeps generally follow a standardized six-sweep blind protocol (C1, C2, C3, L1, M, R1) as shown in Figure 1, though additional sweep types are also present. We created a subset for consistency, with 1,250 patients each containing 6 sweeps. This subset was then divided into 850 / 200 / 200 patients for training, validation, and testing, respectively. For all tasks, we train the downstream AI models using a single sweep from each patient

as one sample. However, the sweep videos of a given patient are not mixed across the train–validation–test sets to prevent data leakage.

Each ultrasound video underwent a uniform preprocessing pipeline prior to model training:

1. removal of embedded textual overlays;

2. spatial cropping to remove machine interface markers while preserving probe orientation markings;

3. resizing of all frames to $224 \times 224$ pixels;

4. normalization to a fixed spatial resolution of $0.75\,\mathrm{mm/pixel}$; and

5. temporal subsampling to 32 equally spaced frames per sweep for consistency across samples.

The processed sweeps were stored in `.pth` format for efficient loading during training.

### 3.2. Experiments Design

Blind-sweep ultrasound acquisition is intended to be simple, but real-world operator variability, especially in settings with limited training, may lead to deviations from recommended protocols. These deviations may include unintended reversal of sweep direction, flipping of the transducer, or premature termination of the sweep. Because such variations do not naturally appear in our test set, we simulate them to systematically study their impact on downstream AI tasks.

We introduce three synthetic perturbations into the test sweeps: (i) sequence reversal, (ii) probe flip, and (iii) incomplete sweep. To approximate realistic variability, perturbations are applied probabilistically: 50% of test sweeps are left unaltered, 30% receive exactly one perturbation, 15% receive two perturbations, and 5% receive all three.

**Perturbation definitions:**

1. **Sequence reversal:** the temporal order of frames is inverted to mimic a sweep performed in the opposite direction from the protocol.

2. **Probe flip:** frames are horizontally flipped to simulate accidental transducer inversion.

3. **Incomplete sweep:** a continuous subsequence of $n$ frames (where $n \in [8, 24]$) starting at frame index $m$ (where $m \in [0, 8]$) is extracted to simulate early termination or incomplete anatomical coverage (Figure 3).

### 3.3. Downstream Tasks

To evaluate the sensitivity of AI models to sweep-quality variations, we assess performance on three relevant tasks:

1. **Sweep-tag classification:** identifying the correct sweep type (C1, C2, C3, L1, M, R1), which is foundational for automated BSOU interpretation pipelines.

2. **Fetal presentation classification:** cephalic vs. non-cephalic.

3. **Placenta location classification:** anterior vs. posterior.

Dedicated models are trained for each task using identical data splits and preprocessing.

### 3.4. Quality-Assessment Models and Correction Simulation

After introducing perturbations, we train QA models to detect whether a sweep has undergone sequence reversal, probe flip, or incomplete coverage. These QA models are intended to approximate real-world feedback mechanisms that could alert operators to re-acquire a sweep when protocol deviations are detected.

To evaluate potential deployment benefits, we simulate a "reacquisition" scenario: when a sweep is flagged as low-quality, synthetic perturbations are removed (representing the user repeating the sweep correctly), and downstream tasks are re-evaluated. This allows us to quantify how quality detection and correction influence overall robustness.

### 3.5. Implementation Details

All models are implemented using a video transformer backbone (MViT) (Li et al., 2022). Training is performed with a batch size of 16 and cross-entropy loss for all classification tasks. Optimization uses Adam with an initial learning rate of $3 \times 10^{-5}$ and a Reduce-on-Plateau scheduler (factor 0.8). Training proceeds for 50 epochs with early stopping based on validation performance. Accuracy and macro-F1 are used as evaluation metrics unless otherwise specified.

Table 1: Performance metrics for downstream tasks, showing accuracy and macro F1 scores at both the sweep and patient levels, when the test set is perturbed.

| Task (Perturbed test set) | Accuracy/F1_macro (sweep level, %) | Accuracy/F1_macro (patient level, %) |
|---|---|---|
| Sweep tags | 29.36 / 28.13 | – |
| Fetal presentation | 66.44 / 56.34 | 75.00 / 67.52 |
| Placenta location | 78.25 / 77.69 | 89.50 / 89.24 |

Table 2: Performance of the QA model for various ultrasound sweep perturbation classification tasks

| Task | Accuracy/F1_macro |
|---|---|
| Sequence reversal | 97.42/97.34 |
| Probe flipping | 99.67/99.24 |
| Incomplete sweep | 88.76/85.34 |

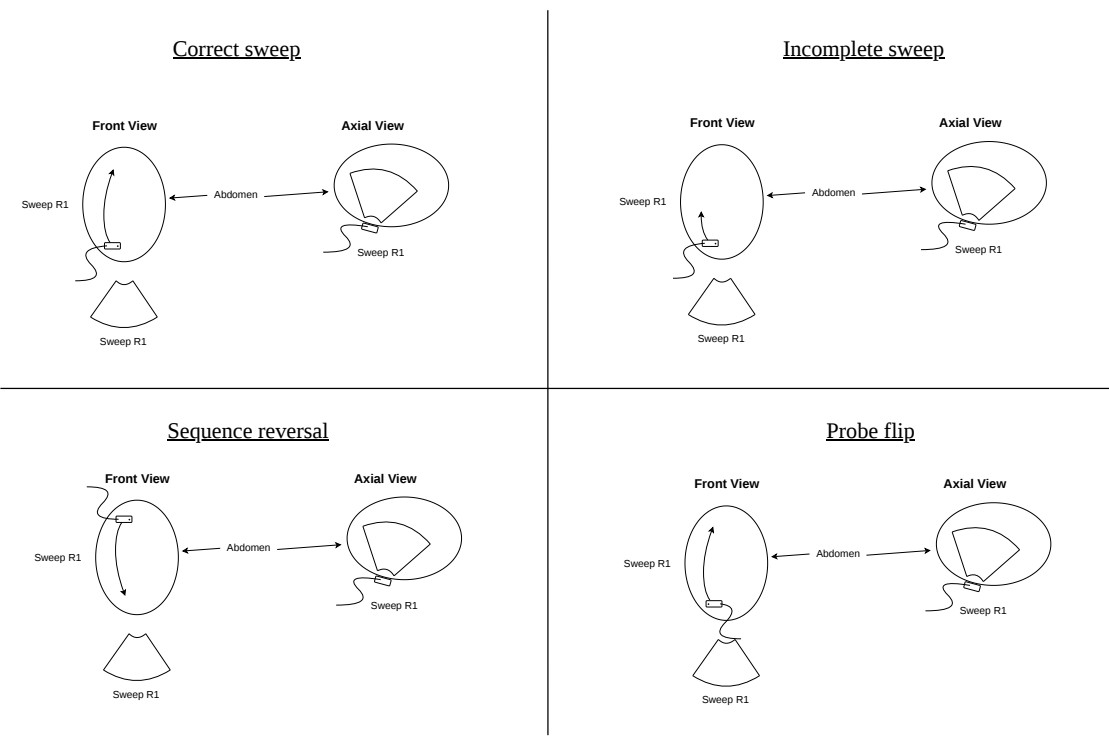

Figure 3: Correct sweep acquisition and three scanning errors - Incomplete sweep, Sequence reversal, and Probe flip

Table 3: Performance of AI models for downstream task after simulated retake of detected perturbed sweep

| Task | Accuracy/F1_macro (sweep level, %) | Δ % | Accuracy/F1_macro (patient level, %) | Δ (%) |
|---|---|---|---|---|
| Sweep tags | 69.79 / 69.98 | 40.43/41.85 | – | – |
| Fetal presentation | 83.75 / 75.40 | 17.31/19.06 | 85.00 / 75.63 | 10.00/8.11 |
| Placenta location | 88.58 / 88.40 | 10.33/10.71 | 93.00 / 92.94 | 3.50/3.70 |

## 4. Results

### 4.1. Performance of AI Models on "Poor-Quality" Test Set

In Table 1, we report model performance on the perturbed test sets across the three downstream tasks. For the Sweep tags task, the model achieves an accuracy of 29.36% only and an F1-macro score of 28.13% at the sweep level. In the Fetal presentation task, performance at the sweep level is 66.44% accuracy and 56.34% F1-macro, while at the patient level, it improves to 75.00% accuracy and 67.52% F1-macro. The Placenta location task shows an accuracy of 78.25% and an F1-macro score of 77.69% at the sweep level, with even higher performance at the patient level, achieving 89.50% accuracy and 89.24% F1-macro.

### 4.2. Performance of Quality-Perturbation Detection Models

Here, we evaluate whether models can accurately detect the presence of these perturbations. As shown in Table 2, the model identifies reversed sequences and flipped probes with 97.42%(97.34%) and 99.67%(99.24%) accuracy(F1_macro), while incomplete sweeps are detected with a accuracy(F1_macro) of 88.76%(85.34%). This capability could be leveraged to provide operators with real-time feedback regarding the type of quality issues in their acquired sweeps.

### 4.3. Simulated Retake and Effect on Downstream Performance

Table 3 shows that downstream model performance on tasks such as sweep tagging, fetal presentation, and placenta location improves once detected perturbations are addressed or filtered by replacing with reacquired sweep data.

- Sweep-tag prediction accuracy and F1-macro increase to 69.79% and 69.98%.

- Fetal presentation increases to 69.79% (accuracy) and 69.98% (F1-macro) at the sweep level, and 85.00% / 75.63% at the patient level.

- Placenta location also increases to 88.58% / 88.40% (sweep level), and 93.00% / 92.94% (patient level).

This improvement in downstream tasks confirms the utility of both artifact recognition and correction in real-world deployment, enabling resilient and trustworthy inference even in the presence of noise during acquisition or operator error.

## 5. Discussion and Conclusion

This work provides an initial examination of how deviations from BSOU acquisition protocols affect downstream AI models and demonstrates that automated QA can help mitigate these effects. Although BSOU protocols are intentionally simple, real-world acquisition, especially by minimally trained operators, may still introduce issues such as reversed sweep direction, probe inversion, or incomplete coverage. These forms of semantic quality variation are seldom represented in existing datasets, leaving open questions about model robustness. Our experiments show that such deviations can meaningfully alter downstream predictions, underscoring the need for explicit QA mechanisms when deploying AI-assisted ultrasound in low-resource settings.

The proposed QA models effectively identify several types of protocol deviations, including sequence reversal and probe flip with high reliability, and incomplete sweeps with reasonable accuracy. This suggests that real-time detection of acquisition inconsistencies is feasible and could serve as an important component of practical BSOU workflows. Our simulated re-acquisition experiment, while simplified, illustrates that correcting flagged sweeps can improve task performance, supporting the idea that operator feedback loops can enhance reliability. The findings complement prior work in other imaging domains, including uncertainty-based quality estimation in echocardiography, which similarly highlights the value of identifying low-quality inputs before downstream analysis.

Several limitations warrant attention. First, we examined only a limited set of perturbations; real-world BSOU quality can also be affected by intensity-based issues such as contact loss, shadowing, and other ultrasound-specific artifacts, which we do not address here. Second, our perturbations are synthetic and may not fully capture the variability of true operator errors. Third, in our simulated workflow, a sweep flagged as low-quality is "re-acquired" by replacing it with a clean version, identical to the original. In practice, repeated sweeps would differ, although the assumption reflects the reasonable expectation that operators would correct their technique once alerted.

In conclusion, this study highlights both the vulnerability of BSOU-based AI models to acquisition variations and the promise of automated QA in supporting more robust and scalable deployment. Integrating real-time quality feedback into BSOU systems may be a key enabler for safe, consistent use of AI-powered obstetric ultrasound in resource-limited environments. Future work should expand QA to broader artifact types, incorporate real re-acquisition data, and evaluate operator–AI interaction in practical field settings.

## Acknowledgments

This work was supported by the Gates Foundation [INV-059090].

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
