# OpenReview forum: "Automated Quality Assessment of Blind Sweep Obstetric Ultrasound for Improved Diagnosis"
_MIDL.io/2026/Conference — MIDL 2026 Poster_

### Official Review · Reviewer_1kLM · 2026-01-09

**Confidence:** 1
**Preliminary Rating:** 5
**Final Rating:** 4

**Summary:**

The manuscript "Automated Quality Assessment of Blind Sweep Obstetric Ultrasound for Improved Diagnosis" addresses a very relevant and interesting topic which can have significant impact on health-care in regions with limited medical resource and consequently often minimally trained medical staff. The investigated quality assessment (QA) model evaluates variations in standardized blind sweep obstetric ultrasound (BSOU) acquisitions and detects perturbations to instantly re-acquire data that sufficed pre-defined quality standards.

**Strengths:**

The problem and goal is very well motivated and nicely introduced. The underlying real-world problem and challenges are clearly stated. The paper focuses much on the application and provides a comprehensive embedding into the problem and task. Chapter 3 (Methods) and 4 (Results) are followed by a fair and critical discussion.

**Weaknesses:**

As this manuscript addresses QA, the results are rather descriptive and less metric-based. (I can not professionally comment on) The three selected perturbations may be too little and limit the overall evaluation of the presented model, however, while I do put this in the weakness section, I do believe that the presented variation fully sufficed for (probably) first research in this field. The main drawback is, that the investigated perturbations are all synthetic and no real-test-data are shown (this is also discussed by the authors).

**Detailed Comments:**

/

**Justification Of Final Rating:**

As indicated me knowledge in this field is limited and my review educated guess only. After reading other reviews I updated to my final rating.
Thanks for the provided discussion items to the remarks.

**Justification Of The Preliminary Rating:**

To my knowledge and in line with the statement of the authors, AI-assisted QA is rather unexplored in BSOU. Thus, a comparison to other/existing methods is not possible or may not be meaningful. Mainly due to the (medical) relevance and the degree of novelty, I suggest the manuscript to be accepted to MIDL.

**Questions To Address In The Rebuttal:**

/

---

> ### Author Response · Authors · 2026-01-25
>
> We sincerely thank the reviewer for their thorough evaluation and strong support.
>
> **Synthetic perturbations** :
>
> We acknowledge that the synthetic perturbation may not fully capture the nuances and detailed variations of real-world acquisitions. Collecting real-world erroneous acquisition data requires prospective data collection or a large scale prospective data where low-quality scans could be detected and sampled, or more realistic synthetic data generation. We agree these are valuable steps, and we are working on these as the next steps as they are non-trivial (difficulty in obtaining ground-truth labels for naturally occuring artifacts) and will be independent study on their own.
>
> While the perturbations we have may be small, what is important is that these perturbations already impact the downstream AI models, and hence are important to be tackled; we provide the first systematic approach to highlight this and study this which is an important first step towards building a robust deployable models in real-world settings.

---

### Official Review · Reviewer_SGLe · 2026-01-10

**Confidence:** 4
**Preliminary Rating:** 3
**Final Rating:** 4

**Summary:**

This paper presents a systematic study of quality variability in Blind Sweep Obstetric Ultrasound (BSOU) and its impact on downstream AI tasks. The authors simulate realistic acquisition deviations (sequence reversal, probe flip, incomplete sweeps), evaluate their effect on multiple clinically relevant tasks, and propose automated quality-assessment (QA) models to detect these perturbations. Experiments show demonstrates that incorporating QA can substantially improve downstream performance.

**Strengths:**

1. The paper is well organized, easy to follow, and appropriately references prior work. Limitations are acknowledged explicitly, which reflects good scientific practice.
2. This QA feedback system is also used in other image-quality aware applications.
3. The results convincingly show that even simple deviations can severely degrade performance, particularly for sweep-tag classification. This finding alone is an important contribution to the BSOU literature.

**Weaknesses:**

1. Synthetic perturbations may not fully capture real-world errors. The simulated deviations may oversimplify operator behavior. Real acquisition errors can be more subtle or more complicated.
2. Simplified re-acquisition assumption may increase total scanning time and might cause clinical issue.

**Detailed Comments:**

The novelty of the paper lies not in proposing a new architecture, but in reframing BSOU reliability as a quality-assurance problem.
There are related papers that use fetal pose estimations for fetal MRI re-acquisitions https://arxiv.org/abs/1907.04500, https://arxiv.org/abs/2007.08146. The authors could add these into introduction.
How frequently the simulated perturbations (sequence reversal, probe flip, incomplete sweeps) occur in real BSOU deployments?
Is video VIT fine-tuned or trained from scratch? Can authors provide more details?
As I mentioned before, please discuss inference-time overhead.

**Justification Of Final Rating:**

I'm happy with what authors responded to my concerns and questions.
I was originally concerned about the synthetic experiments but authors convinced me with "aim to open directions for addressing more subtle, operator-dependent BSOU quality issues.".
"Importantly, the proposed framework does not mandate automatic reacquisition for all QA failures. Instead, it supports informed decision-making by operators or clinicians. We view this as a conservative, safety-oriented abstraction intended to illustrate the potential role of QA feedback, rather than a prescriptive clinical protocol."
The pipeline and idea can play an important role in the real clinical application with more future studies.
Thank the authors for the rebuttals.
I raised my rating to accept.

**Justification Of The Preliminary Rating:**

The rating is based on the paper’s strong clinical relevance, sound experimental design, and clear contribution to the reliability of AI-assisted obstetric ultrasound. Only using synthetic perturbations and lack of discussions of inference time overhead limit the quality of the paper.

**Questions To Address In The Rebuttal:**

As stated in the section above, I'd like to hear more from the authors.

---

> ### Author Response · Authors · 2026-01-25
>
> We thank the reviewer for their thoughtful evaluation and constructive feedback. We appreciate the recognition of our paper’s clinical relevance and experimental design. We address the specific questions and concerns below.
>
> **Synthetic Perturbations and frequency in Real Deployments:**
> We have addressed this concern in the common response at the beginning.
>
> **Re-acquisition Time and Clinical Workflow:**
> We acknowledge that reacquisition may increase total scanning time in some cases. Based on real-world experience, a typical scan takes approximately 2 minutes, and if reacquisition is required, it takes an additional 1 minute of processing time to determine whether reacquisition is necessary. Notably, in current BSOU workflows, failed or invalid acquisitions already require repeat scanning once identified, often after downstream review.
>
> Our QA-based approach is designed to identify such failures earlier and provide immediate feedback, which has the potential to reduce wasted scans and overall workflow inefficiency rather than increase it.
>
> Importantly, the proposed framework does not mandate automatic reacquisition for all QA failures. Instead, it supports informed decision-making by operators or clinicians. We view this as a conservative, safety-oriented abstraction intended to illustrate the potential role of QA feedback, rather than a prescriptive clinical protocol.
>
> **ViT Implementation and Inference Details:**
> The video ViT backbone is fine-tuned from weights pre-trained on the KINETICS dataset. Inference-time overhead is a few seconds on the CPU. For resource-constrained deployment, the QA model can be distilled to operate in low resources.
>
> **Suggested References on Fetal Pose Estimation:**
> We thank the reviewer for suggesting papers related to fetal MRI pose detection (Xu et al., 2019, arXiv:1907.04500; Zhang et al., 2020, arXiv:2007.08146). We have included these works as references in the related works section.

---

### Official Review · Reviewer_SggF · 2026-01-15

**Confidence:** 3
**Preliminary Rating:** 1
**Final Rating:** 3

**Summary:**

This paper studies the impact of acquisition-quality deviations in Blind Sweep Obstetric Ultrasound (BSOU) on downstream AI tasks and explores automated quality-assessment (QA) models to detect such deviations. While the problem is clinically relevant and timely, the current submission has substantial issues in completeness, methodological contribution, and experimental validity. In its present form, the paper does not meet the standards of a full MIDL submission.

**Strengths:**

The paper addresses a clinically relevant and timely problem. The study design clearly illustrates how acquisition deviations can impact multiple downstream AI tasks, which may be informative for future robustness analyses and system deployment considerations.

**Weaknesses:**

1. The paper contains multiple unresolved placeholders in critical sections (e.g., “XXX” in Section 3.1), including dataset sources, sample sizes, and cohort descriptions. These are essential details required to assess data validity, experimental rigor, and reproducibility. The presence of such placeholders indicates that the manuscript is unfinished and prevents meaningful evaluation. This issue alone is sufficient grounds for rejection.

2. The paper does not introduce a novel QA methodology or learning formulation. The core approach consists of training standard video transformer classifiers to detect simple, synthetically introduced perturbations. While the empirical analysis may be informative as a robustness study, it lacks methodological novelty expected of a full MIDL paper and reads more like an engineering validation or exploratory analysis rather than a research contribution.

3. The simulated perturbations are binary and easily detectable, which likely explains the near-ceiling QA performance reported. More subtle and clinically realistic BSOU quality issues such as incomplete signal dropout, shadowing, or motion artifacts are not addressed. As a result, the proposed QA setting does not convincingly reflect the real-world difficulty of BSOU quality assessment, limiting the practical and scientific relevance of the findings.

4. The related work section is organized primarily by imaging modality (MRI, X-ray, mammography, ultrasound). For a BSOU-focused paper, a more appropriate structure would categorize prior work by QA strategy and explicitly contrast BSOU with other ultrasound and non-ultrasound settings. The current framing obscures how this work advances the BSOU QA literature.

**Detailed Comments:**

N/A

**Justification Of Final Rating:**

The clarification regarding the placeholders and anonymization addresses concerns about the manuscript's completeness, and the authors’ responsiveness and update to the related work section are appreciated. As stated by the other reviewers, the clinical motivation of the paper is clear, the dataset size is large, and it matches the interests of MIDL. My core concern is the paper's external validity, as the disturbances provided in the experiments are all synthetic, and the settings seem to be too simple and straightforward. I doubt whether these simulations can represent real-world cases. Sequence reversal and horizontal flips are very strong geometric signals in videos, which can lead to artificially high accuracy. Based on these considerations, I updated my recommendation.

**Justification Of The Preliminary Rating:**

While the topic is important and relevant, the manuscript is incomplete and lacks sufficient methodological depth and realism to justify acceptance as a full MIDL paper. Substantial revision and completion would be required before it could be reconsidered.

**Questions To Address In The Rebuttal:**

See Weaknesses.

---

> ### Author Response · Authors · 2026-01-25
>
> We sincerely thank the reviewer for their thorough evaluation and detailed feedback. We address each concern below and outline concrete revisions.
>
> **Manuscript Completeness and Placeholders:**
> The placeholders (e.g., “XXX”) were used exclusively for anonymization, under the assumption of a double-blind review process (which we realized after the submission deadline that it was single blind and anonymization was not necessary), and do not indicate missing or incomplete information for reviewing the value of the contribution.  Specifically, the study uses BSOU videos from two clinical sites (Chapel Hill, NC, USA, and Lusaka, Zambia), comprising over 8,000 studies from more than 4,600 pregnant individuals. For consistency, a subset of 1,250 patients with exactly six standardized sweeps was used for analysis. All dataset sources, sample sizes, and cohort descriptions were fully defined prior to submission (except for the names) and are now explicitly stated in the revised manuscript. We regret any confusion caused and emphasize that this does not affect the validity, rigor, or reproducibility of the work.
>
> **Methodological Contribution and Novelty:**
> We respectfully disagree with the reviewer that the methodological novelty is expected of a full MIDL paper, and validation and exploratory analysis not being worthy of MIDL submission. As outlined in the MIDL 2026 call for papers, the conference aims to “foster both algorithmic innovation and clinical translation,” and invites contributions that range “from foundational methodologies to validation studies demonstrating real-world impact.” In addition, the topics of interest include
> “clinical integration, workflow optimization, and real-world deployment.”
>
> In this context, while we employ established video transformer architectures, the novelty of our work lies in highlighting the need for quality assessment in taking BSOU-based AI systems into clinical reality which has been lacking in the literature. We then further provide novel study design and framework to study this sytematically, by focusing on common semantic low-quality data acquisition while showing how this impacts the models deployed by training them only on curated high-quality videos. These insights are built from our own ongoing research projects where we have collected nearly 5000 videos from across five different countries, our experience in working towards clinical translational work which requires regulatory approvals where this sort of quality assurance is an important step.
> We systematically define realistic operator-induced perturbations encountered in low-resource settings, and evaluate their effects across multiple downstream tasks within a unified framework.
>
> This goes beyond an exploratory robustness analysis by providing a task-agnostic QA perspective and demonstrating that automated QA with operator feedback can consistently improve downstream performance. By addressing a key barrier to safe and scalable clinical deployment of BSOU systems, our work contributes methodological insight at the level of problem formulation, evaluation protocol, and clinical relevance, aligning with MIDL’s scope of including clinical translational work.
>
> **Realism of Perturbations:**
> We have addressed some part of this concern in the common response given at the beginning. For the remaining part, we have addressed it below. Although the selected perturbations are binary in nature, they can have a substantial impact
> on downstream tasks, as highlighted by Reviewer SggF, highlighting the importance of detecting them. To our knowledge, no prior work has addressed similar QA settings (as also noted by Reviewer 1kLM), and we believe that our study provides an important first step towards comprehensive QA framework for AI systems taking BSOU videos as input.
>
> Regarding the concern about shadows, for BSOU shadows should not be considered as low-quality (due to non-compliance in BSOU protocol) if the shadows are due to maternal and fetal anatomy wrt the probe positions. This is because BSOU by design takes blind sweeps, focusing on predefined trajectory in women's abdomen irrespective of what video content is captured. Thus, shadows in this case would be natural expected variation in the dataset that the AI models should be robust against. Our focus is in low-quality data due to protocol non-compliance.
>
> **Related Works:**
> We thank the reviewer for the feedback on the Related Works section. We agree that the suggested structure is more appropriate and have updated the Related Works section of the manuscript accordingly.

---

### Author Rebuttal · Authors · 2026-01-25

**Rebuttal:**

We sincerely thank the reviewers for their thoughtful evaluation and constructive feedback. We appreciate the recognition of the **clinical relevance and timeliness** of our work (Reviewer **SggF**), the acknowledgment that the paper is **well organized, easy to follow, and appropriately referenced** (Reviewer **SGLe**), and the appreciation for our **clear motivation, real-world grounding, and fair discussion** (Reviewer **1kLM**). We also thank Reviewer **1kLM** for noting that AI-assisted QA in BSOU is largely unexplored, underscoring the **novelty** of our study.

Our study focuses on **systematic investigation of BSOU data-quality issues** and their impact on downstream AI tasks an aspect Reviewer **SggF** highlighted as important for future robustness analyses and deployment. Despite BSOU’s growing use, acquisition-quality deviations are often overlooked, and real-time mechanisms to detect and mitigate them remain absent.

Although the perturbations we examine—sequence reversal, probe flip, and incomplete sweeps—are simple, our findings (noted by Reviewers **SGLe** and **1kLM**) show that they can **severely degrade downstream performance**, particularly for sweep-tag classification. Reviewer **SGLe** emphasized that this result alone is an important contribution to BSOU literature and aligns with QA needs in other image-quality-aware applications.

We also aim to open directions for addressing more subtle, operator-dependent BSOU quality issues. Reviewer **1kLM** appreciated the comprehensive framing of the problem and task, and future work will extend this to pixel-level artifacts like motion or dropout.

All reviewers raised valid concerns about relying on synthetic perturbations. While we agree on this limitation, collecting real datasets with controlled acquisition errors is currently impractical due to limited data, high cost, and clinical staffing requirements. Reviewer **SGLe** noted our explicit acknowledgment of limitations as good scientific practice.

Below, we address reviewers’ specific questions and concerns.

**Supporting Material:**

/attachment/99d4eaaba3bd91fbf82bb5e1dbff6c411d9289b4.pdf

---

### Comment · Area_Chair_WosK · 2026-01-25
**The paper is now open for discussion**

Dear Reviewers,

The authors have submitted their responses to the comments you raised. The paper is now open for discussion.

Please engage with the authors during this period to clarify any remaining issues.

---

### Meta-Review · Area_Chair_WosK · 2026-02-06

**Recommendation:** Accept (Poster)
**Confidence:** 5

**Metareview:**

Variations in sweep quality during ultrasound image acquisition—particularly in obstetrics and in data collected by novice users—represent an important challenge and a strong motivation for this work. The perturbations introduced, although synthetic, are reasonably representative of realistic scenarios that may be encountered in practice. I believe the study will spark valuable discussion on how such approaches can be further improved.

While the technical novelty is limited, the application focus and the investigation of failure cases in ultrasound data collection are both interesting and relevant. Overall, the work has merit and should be accepted following rebuttal.

---

### Decision · Program_Chairs · 2026-02-14

Accept (Poster)